# Precision Control of Spraying Quantity Based on Linear Active Disturbance Rejection Control Method

**Xin Ji** [1,2] , **Aichen Wang** [1,2] **and Xinhua Wei** [1,2,*]

1 College of Agricultural Engineering, Jiangsu University, Zhenjiang 212013, China; 2112016020@stmail.ujs.edu.cn (X.J.); acwang@ujs.edu.cn (A.W.)
2 Key Laboratory of Modern Agricultural Equipment and Technology, Ministry of Education, Zhenjiang 212013, China
* Correspondence: wxh@ujs.edu.cn; Tel.: +86-0511-8879-6996

**Abstract:** Current methods to control the spraying quantity present several disadvantages, such as poor precision, a long adjustment time, and serious environmental pollution. In this paper, the flow control valve and the linear active disturbance controller (LADRC) were used to control the spraying quantity. Due to the disturbance characteristics in the spraying pipeline during the actual operation, the total disturbance was observed by a linear extended state observer (LESO). A 12 m commercial boom sprayer was used to carry out practical field operation tests after relevant intelligent transformation. The experimental results showed that the LADRC controller adopted in this paper can significantly suppress the disturbance in practical operation under three different operating speeds. Compared with the traditional proportional–integral–differential controller (PID) and an improved PID controller, the response speed of the proposed controller improved by approximately 3~5 s, and the steady-state error accuracy improved by approximately 2~9%.

**Keywords:** precision agriculture; spraying quantity control; linear active disturbance rejection; disturbance observer; variable spraying control

## 1. Introduction

Pesticide spraying is still the main means of pest control in agricultural production in many countries. Precision spraying technology can greatly improve the effective utilization rate of pesticides and meet the current demand for green agriculture [1,2]. However, due to the complex field operation environment, the acceleration and deceleration of vehicles and other factors in the actual process will cause the disturbance of pressure and flow inside the pipeline. At present, there are related commercial products for spraying quantity control. For example, Radion 8140 (Teejet, IL, USA), PW-KZ1 (HuaiYu, Changzhou, China), and LC-19 (LiCheng, Ningbo, China) are the main spraying quantity control systems that are widely adopted in China. Based on the feedback of actual usage, the control results are in the range of 5~15% [3]. Therefore, it has become an important research direction in the field of plant protection to develop a precise variable spraying control system.

Many scholars have carried out mechanism analysis and modeling of spraying quantity regulation systems with different principles. They developed relevant controllers and also achieved certain research results. Four control methods are available to be widely used in agricultural spraying control systems and related fields: PID control [4], intelligent control [5], fuzzy control [6–8] and neural network control [9]. Guzman et al. [4] obtained the transfer function of the proportional flow valve in the main pipe through the reaction curve method. According to the nonlinear characteristics of the flow valve, the PID control parameters were adjusted by the loop shaping method. This method was robust to a certain extent, but it was difficult to find the multiple control parameters. Felizardo et al. [5] constructed the state-space model of a spraying system and established the optimal quadratic index cost function through control input and the steady-state error. By adjusting

the amount of weight matrix in order to obtain the optimal control, the steady-state error control was limited within 5%, but the cost function only considered the control input and the steady-state error, hence system response performance was difficult to be guaranteed. Shi and Liu et al. [6,7] developed electromechanical control valves, respectively, and modeled them by mechanism analysis. They adopted a fuzzy PID control algorithm in order to control the spraying quantity. The fuzzy PID controller was significantly better than the traditional PID in response time and overshoot index, but the disturbance was not taken into account by the authors. Song et al. [8] used an adaptive fuzzy controller to control the electromechanical control valve. The system could realize online tuning of control parameters and the control algorithm had strong robustness. Wang et al. [9] built a multi-sensor operation parameter monitoring system and used a neural network to carry out self-learning of the PID control in order to ensure the operation effect under different speeds and target quantity. However, the system had certain requirements on the computing ability for actual field operations, so there are some problems in its large-scale promotion. Wei et al. [10] found that the diaphragm pump would produce a periodic pressure pulsation phenomenon, and the pulsation period and amplitude would periodically change with the speed of the diaphragm pump. This phenomenon will have a certain influence on the internal flow of pipelines. To sum up, current studies on spraying quantity control mainly focus on the response characteristics of the control system and the self-tuning of the controller parameters. The PID controller, which is dependent on the linear combination of the error, is mainly adopted in the spraying quantity control. It has two main disadvantages. First, the related control methods often ignore the disturbance of the flow and pressure in the pipeline caused by the vehicle speed and so on. The acceleration and deceleration of the vehicle will cause the change in the speed of the diaphragm pump, which will affect the internal flow and the pressure in the pipeline. At the same time, the response curve method and least square fitting method are used in the modeling process of the controlled plant, which has certain parameter perturbation. Second, the PID controller has a problem in that the control parameters are difficult to tune. Hence, further research is still needed for the spraying control system with nonlinear, time-varying, hysteresis, and disturbance characteristics.

It is undeniable that the combination of intelligent algorithms is a good solution for model uncertainty and controller parameter tuning. The neural network, fuzzy logic method, and other methods have the ability of model approximation and self-adaptation, and they can play the role of nonlinear online approximation and compensation. Intelligent algorithms are already used in some industrial applications [4–9]. In the case of model uncertainty that exists in spraying quantity control systems, further research is needed on how to effectively use the model information and the intelligent methods to realize the online identification of flow valve model parameters. In addition, the intelligent methods can also reduce the difficulty of the controller parameter tuning. In the literature [4–9], they all adopted intelligent algorithms in order to find the best controller parameters.

In this study, the perturbation of the flow valve model parameters and external disturbance that occurred in the actual operation of the applicator was regarded as the total disturbance by the extended state observer, and the disturbance was compensated in real-time by the controller in order to achieve the purpose of restraining the disturbance. Considering the steady-state error, adjustment time, and overshoot index comprehensively, a cost function was constructed, and the controller parameters were optimized by using the particle swarm optimization (PSO) algorithm. The second-order linear active disturbance rejection control algorithm was verified by simulation and field experiments.

## 2. Modeling and Problem Description

### 2.1. Flow Valve Structure and Transfer Function Model

In this study, the flow control valve, as shown in Figure 1, was used as the control plant to regulate the flow of liquid in the main pipe. The flow valve is composed of a DC motor, gear train, screw, spool, etc. The flow valve contains an input port, an output port,

and an overflow port. The displacement of the spool is changed by controlling the rotation of the DC motor with the size of the overflow port opening changing.

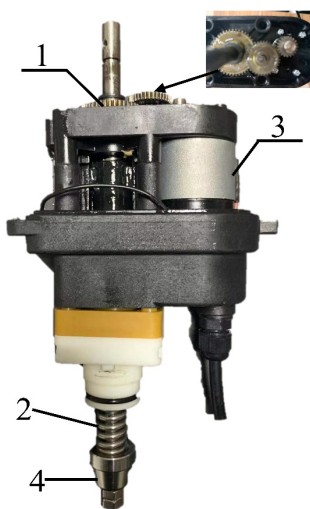

**Figure 1.** Structure of flow control valve: 1, gear train; 2, screw; 3, DC motor; 4, spool valve.

The flow valve is composed of the valve body and the control mechanism. The mechanical inertia and electromagnetic inertia of the DC motor in the valve were ignored, and the delay characteristic of the flow control object was considered. The controlled object is simplified into the first-order inertia with a delay link [11], whose transfer function is

$$G(S) = \frac{K_v}{(T_v S + 1)} e^{-\tau s} \tag{1}$$

where $K_v$ denotes the gain of the controlled plant; $T_v$ is the time constant of the first-order inertia link; $\tau$ represents the delay time constant; $S$ is the Laplace operator. The response curve method was used to identify the parameters of the controlled object. Flow data were collected through a data acquisition card [4,12]. The first-order inertia time constant of the controlled object was 4.6 s, and the delay time constant was 0.8 s. The gain of the controlled plant was 71.4. To simplify, the delay link can be regarded as a first-order inertia link, and then the transfer function of the controlled plant can be approximated as

$$G(s) \approx \frac{K_v}{(T_v S + 1)(\tau S + 1)} = \frac{20.41}{S^2 + 1.63S + 0.29} \tag{2}$$

**Remark 1.** *The transfer function model of the controlled plant in Equation (2) is the nominal system model obtained through the response curve method. There is certain parameter perturbation in the actual system. This part can be regarded as the internal disturbance of the system (uncertainty). To our knowledge, the researchers mostly adopt PID and other methods, and the research mainly focuses on PID controller parameter tuning, considering the disturbance in the actual operation less; however, this disturbance cannot be ignored and has an important impact on the operation effect. The common form of PID control is $u = K_p e(t) + K_i \int_0^t e(\tau) d\tau + K_d \frac{de(t)}{dt}$; where $K_p, K_i, K_d$ are the controller parameters.*

### 2.2. Problem Description and Related Definitions

For the convenience of expression, we briefly explain some symbols, definitions, and lemmas used in the following paragraphs.

**Definition 1** ([13]). *Considering a single-input second-order linear uncertain plant*

$$\ddot{y}(t) = -(A_n \pm \Delta A)\dot{y}(t) - (B_n \pm \Delta B)y(t) + (C_n \pm \Delta C)u(t) + \omega(t) \tag{3}$$

*where $y(t)$ is the output, $y(t) \in \mathbb{R}$; $u(t)$ is the control input, $u(t) \in \mathbb{R}$; $A_n, B_n, C_n$ are the nominal plant parameters; $\Delta A, \Delta B, \Delta C$ are the unknown model uncertainties introduced by the plant parameters, nonlinear friction, and unmodeled dynamics; $w(t)$ denotes uncertain external disturbance. The second-order plant model can be rearranged as*

$$\ddot{y}(t) = -A_n \dot{y}(t) - B_n y(t) + C_n u(t) + f \tag{4}$$

*where $f$ denotes the lump disturbance that is given*

$$f = \pm \Delta A \dot{y}(t) \pm \Delta B y(t) \pm \Delta C u(t) + \omega(t) \tag{5}$$

Without loss of generality, we can obtain the higher dimensional definitions.

**Definition 2** ([13])**.** *Considering the following system with disturbance and uncertainty*

$$y^{(n)} = h\left(y, y^{(1)}, y^{(1)}, \cdots, y^{(n-1)}\right) + \omega(t) + bu \tag{6}$$

*where $h$ contains unmodeled error; $w(t)$ denotes external disturbance; $u, y$ represents input and output, respectively.*

**Definition 3** ([13])**.** *Choosing $[x_1, \cdots, x_n]^T = \left[y, \cdots, y^{(n-1)}\right]^T$ as the state variable, $f$ represents the lump disturbance, which contains internal disturbance and external disturbance, and $f = -a_1 \frac{d^{n-1}y}{dt} - a_2 \frac{d^{n-2}y}{dt} - \cdots - a_n y + \omega$, and $x_{n+1} = f$ as the extended state variable, we can obtain the extended state-space description*

$$\begin{cases} \dot{x}_1 = x_2 \\ \dot{x}_2 = x_3 \\ \quad \cdots \\ \dot{x}_n = x_{n+1} + bu \\ \dot{x}_{n+1} = \dot{f} \\ y = x_1 \end{cases} \tag{7}$$

**Assumption 1.** *The lump disturbance of the System (6) and its derivatives are bounded.*

**Lemma 1** ([13])**.** *According to System (6), a linear extended state observer (LESO) is constructed in the following form:*

$$\begin{cases} \dot{\hat{x}}_1 = \beta_1(y - \hat{x}_1) + \hat{x}_2 \\ \dot{\hat{x}}_2 = \beta_2(y - \hat{x}_1) + \hat{x}_3 \\ \quad \cdots \\ \dot{\hat{x}}_n = \beta_n(y - \hat{x}_1) + \hat{x}_{n+1} + bu \\ \dot{\hat{x}}_{n+1} = \beta_{n+1}(y - \hat{x}_1) \end{cases} \tag{8}$$

*where $[\beta_1, \beta_2, \cdots, \beta_{n+1}]$ are the observer gains; $\hat{x}_i (i = 1, 2, \cdots, n+1)$ is the estimated value of the state.*

**Lemma 2** ([13])**.** *The state feedback controller of (6) is designed for the decoupled integral series System (7)*

$$u = \frac{1}{b_0}(l_n r - l_n \hat{x}_1 - l_{n-1}\hat{x}_2 - \cdots l_1 \hat{x}_n - \hat{x}_{n-1}), \tag{9}$$

*where $[l_1, l_2, \cdots, l_n]$ are controller gains; $r$ denotes the reference input, then the system converges to the equilibrium point asymptotically.*

**Remark 2.** *The core idea of LADRC is to conduct disturbance observation through the extended state observer and introduce the lump disturbance value obtained by observation into the control*



*channel for compensation. It can achieve the effect of disturbance suppression [13–17]. The PID controller mainly relies on an integral action in order to suppress the disturbance and has a certain ability to suppress the constant disturbance. Relevant literature has made a detailed analysis of its robustness. In addition, sliding mode control, robust control, and adaptive control have certain effects on disturbance and model parameter perturbation. The LADRC controller that is adopted in this study was essentially a two degree of freedom controller of "PD controller + observer", which is easy to be implemented in engineering and has a large number of application cases.*

### 2.3. System Working Principle and Hardware Design

In the variable spraying operation, the prescription value is obtained according to the geographic location and other information; that is the target spraying quantity in the current operating area. In the process of the operation, the speed of the vehicle is monitored by a sensor and the real-time spray volume and speed are matched by adjusting the liquid flow in the pipeline. The variable spraying control system belongs to the type of follow-up control; that is, the target value of the liquid flow in the pipeline changes with the speed and other information [18,19]. The formula for calculating the liquid flow of the target spraying quantity is

$$Q_{rel} = \frac{vL\varphi(x,y)}{600} \tag{10}$$

where $Q_{rel}$ is the target spraying quantity (L/min); $v$ denotes the speed of the vehicle (km h$^{-1}$); $\varphi(x,y)$ represents the prescription value (L hm$^{-2}$); $L$ denotes the length of the sprayer boom (m).

The spraying system is mainly composed of a chemical tank, a diaphragm pump, a pressure stabilizing package, a multistage filter, a zoning valve, and a safety valve, which is shown in Figure 2. The driving shaft of the vehicle is used to provide power and the liquid in the chemical tank is sent into the pipeline at a good rate according to the agronomic requirements. To prevent excessive pressure caused by pipeline blockage and other factors, tightening the safety valve should be adjusted before operation. During operation, when the operating pressure of the pipeline exceeds the preset value, the relief port of the safety valve will be opened and the liquid will flow back into the chemical tank. According to the number of spray nozzles, a certain number of partition valves are added in order to realize the electric control of the spray nozzles. Before the operation, the spray nozzle can be selected manually or by touch screen control mode, and the spraying area can be selected in a specific operational area. In this study, the 3WPF400 sprayer (Essen Agricultural Machinery Co., Ltd., Changzhou, China) was carried out using intelligent modification, and the positioning system, the prescription map decision-making interpretation system, spraying control system, etc., were installed, respectively. The upper data acquisition computer built by LabVIEW was used to carry out real-time sampling of the spraying operation parameter information in the actual operation process. The 32-bit chip STM32F107 (TI Corporation, Dallas, TX, USA) was adopted as the core controller. The onboard resources include five drive outputs and one CAN2.0 communication, an isolating RS485/232 communication, three isolating 12-bit high-precision voltage ADC sampling channels, two isolating optocoupler pulses capture channels, and one adjustable PWM output channel. In the spray main pipeline, the pressure withstands value was 0~2 MPa and the flow range was 0~90 L/min. A QCT2019 pressure sensor and QCWG2 turbine flow sensor (Tianyu Hengchuang Technology Co., Ltd., Beijing, China) were selected. The experimental prototype is shown in Figure 3.

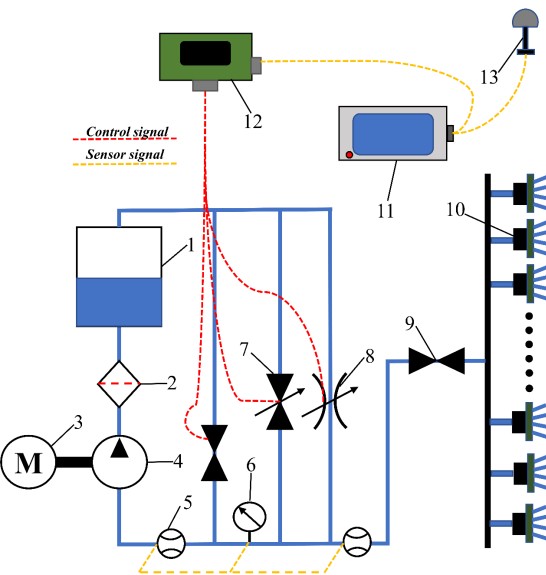

**Figure 2.** Structure of spraying system: 1, tank; 2, filter; 3, power source; 4, diaphragm pump; 5, flow sensor; 6, pressure sensor; 7, safety valve; 8, flow regulating valve; 9, divider valve; 10, nozzle; 11, onboard computer; 12, spray controller; 13, DGPS.

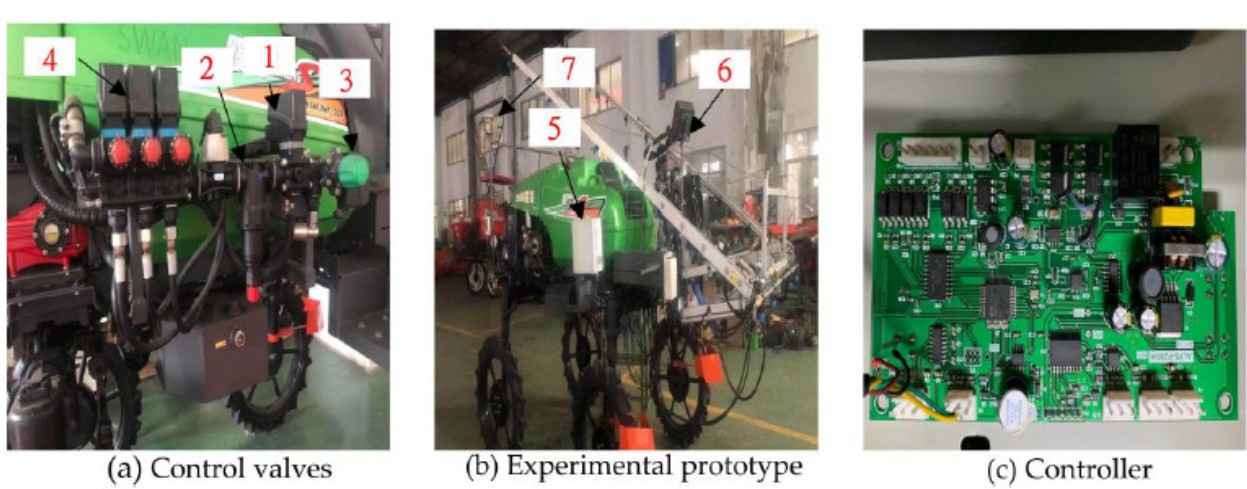

**Figure 3.** Field experiment prototype: 1, flow valve; 2, filter; 3, relief valve; 4, partition valve; 5, controller box; 6, computer; 7, GPS speed sensor.

## 3. Controller Design

### 3.1. Design of Spraying Controller

In the field operation of the sprayer, due to change in speed, there is a change in the diaphragm pump speed, thus generating a certain flow and pressure fluctuation in the pipeline. In the modeling process of the flow control valve, the internal disturbance will occur due to the uncertainty of the relevant object parameters. Due to the disturbance in the field operation, based on the ADRC technology, this study constructed a LESO, which regards the external disturbance, modeling error, and internal disturbance as the lump disturbance [14]. The lump disturbance is observed by the LESO and the real-time disturbance closed-loop compensation is carried out. Based on the decoupling characteristics of the ADRC technology, the original system is converted into a series integrator and the state feedback controller is designed [13–17]. The control structure is shown in Figure 4.

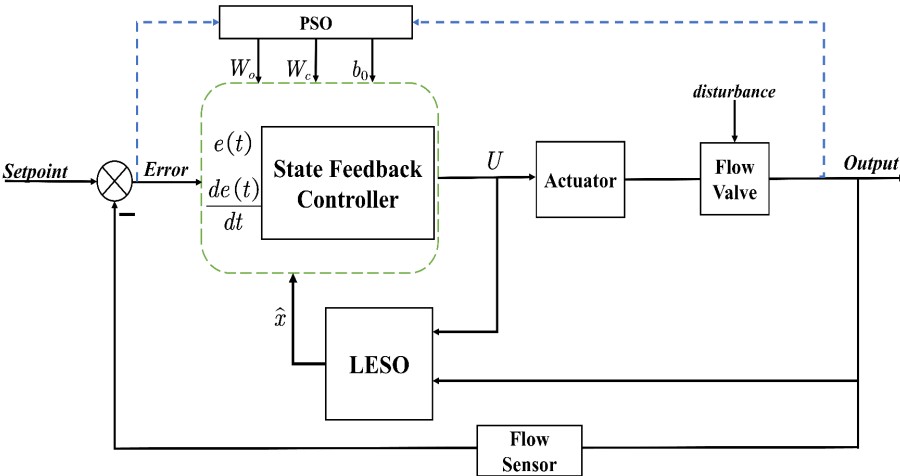

**Figure 4.** Structure of LADRC.

According to Lemma 1 and 2, the state-space model, linear extended state observer, and controller of the system are established, and they are rewritten into the form of a state-space equation as follows:

$$\begin{cases} \dot{\hat{x}}_1 = -\beta_1\hat{x}_1 + \hat{x}_2 + \beta_1 y \\ \dot{\hat{x}}_2 = -(\beta_2 + l_2)\hat{x}_1 - l_1\hat{x}_2 + \beta_2 y + l_2 r \\ \dot{\hat{x}}_3 = -\beta_3\hat{x}_1 + \beta_3 y \\ u = \frac{1}{b_0}(l_2 r - l_2\hat{x}_1 - l_1\hat{x}_1 - \hat{x}_3) \end{cases} \tag{11}$$

The poles of the observer and the controller are, respectively, allocated at the observer bandwidth and the controller bandwidth [17] through the pole allocation method, and the relevant gain parameters are obtained as

$$\begin{cases} l_1 = 2\omega_c, l_2 = \omega_c^2 \\ \beta_1 = 3\omega_0, \beta_2 = 3\omega_0^2, \beta_3 = \omega_0^3 \end{cases} \tag{12}$$

where $\omega_0, \omega_c$ denotes the observer bandwidth and controller bandwidth, respectively.

**Remark 3.** *In essence, the second-order LADRC can be equivalent to "a second-order low-pass filter + a PID controller"; relevant literature has given theoretical proof and you can look up the paper of Tian for details [13]. Compared with the traditional "PID + filter" form, LADRC has the characteristics of easier parameter tuning and stronger anti-interference ability.*

### 3.2. Optimal Controller Parameter Tuning Algorithm

To solve the problem that the control parameters of the traditional controller are difficult to tune, the particle swarm optimization algorithm (PSO) was adopted in this paper in order to optimize the controller parameters [20]. The core idea of PSO is to optimize parameters by updating the local optimal value, current global optimal value, particle position, and velocity information through a certain number of particles flying at a certain speed in a specified dimensional space [20]. First, it is assumed that $N$ is the dimensional search space and $M$ is the particle swarm size. Then we can define the current position of a particle $X_i(t) = [X_{i,1}, X_{i,2}, \cdots, X_{i,N}(t)]$.

The cost function is constructed as follows:

$$argminF = -\alpha_1\left(\frac{t_s}{5 \cdot 10^{-2}} + 1\right) - \alpha_2\ln\left(\frac{\sigma}{10^{-4}} + 1\right) + \alpha_3\int_0^\infty |e(t)|dt \tag{13}$$

where $t_s$ is the steady-state adjustment time; $\sigma$ is the overshoot; $\int_0^\infty |e(t)|dt$ is the absolute error integral (ITAE) of the deviation between the reference input and the actual output; $\alpha_1, \alpha_2, \alpha_3$ are inertia weight coefficients, respectively, and $\alpha_1 + \alpha_2 + \alpha_3 = 1$.

$X_i(t)$ is substituted into the cost function in order to find the cost value, and then we can define the current velocity of a particle $V_i(t) = [V_{i,1}(t), V_{i,2}(t), \cdots, V_{i,N}(t)]$. The best position of the current particle can be obtained and defined as $P_i(t) = [P_{i,1}(t), P_{i,2}(t), \cdots, P_{i,N}(t)]$.

Then the cost function value of the current position of the $i$ particle is compared with the fitness value of the best position of the individual. If it is greater than the latter, then $P_i(t) = X_i(t)$, otherwise $P_i(t) = P_i(t-1)$ [20].

The global best position is the current position with the best cost function value found by PSO. It can be described as $G(t) = P_s(t) = \left[P_{g,1}(t), P_{g,2}(t), \cdots, P_{g,N}(t)\right]$ where $1 \leq g \leq M, g = argmin\{F\}$ [20].

The updating formulas of particle speed and position are as follows:

$$V_{i,j} = WV_{i,j}(t) + \alpha \cdot rand1 \cdot \left(P_{i,j}(t) - X_{i,j}(t)\right) + \beta \cdot rand2 \cdot \left(G_{i,j}(t) - X_{i,j}(t)\right), \quad (14)$$

$$X_{i,j}(t+1) = V_{i,j}(t) + X_{i,j}(t) \quad (15)$$

where $1 \leq i \leq M, 1 \leq j \leq N$; $\alpha, \beta$ denotes the acceleration coefficients; $rand1, rand2$ is the uniformly distributed sequence of independent random numbers with a value range of $(0,1)$, respectively; $W$ represents the inertia weight coefficient. We adopted the linear decrease formula in order to adjust the search scope. It can be described as follows:

$$W = W_{start} - \frac{W_{start} - W_{end}}{t_{max}} \quad (16)$$

where $t$ denotes the number of current iterations; $t_{max}$ is the maximum number of iterations; $W_{start}, W_{end}$ represents initial inertia weight and termination inertia weight, respectively. The algorithm flow chart is shown in Figure 5.

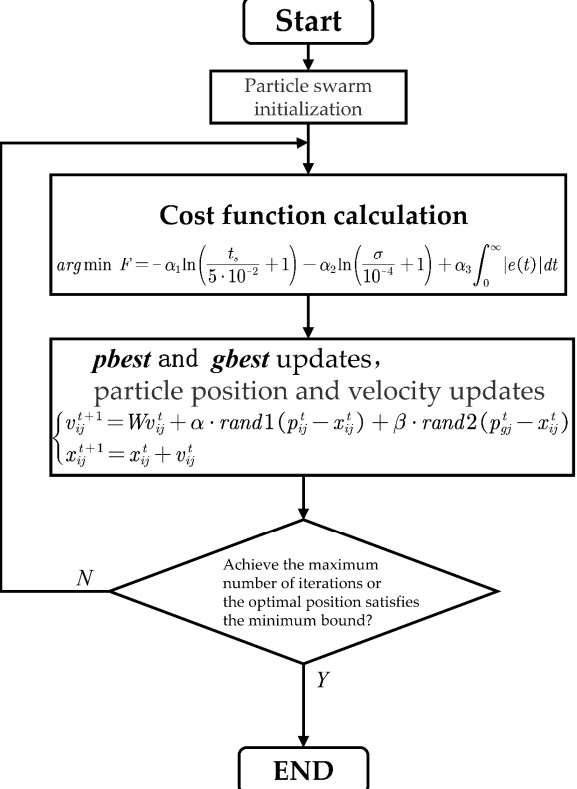

**Figure 5.** Flow chart of PSO parameter optimization.

### 3.3. Controller Comparison and Analysis

To solve the problem of spraying quantity control, many methods have been discussed in [4–8]. The most common method is the PID controller and its improved form. Similar to the spraying quantity control system, it has different characteristics and requirements in different stages. In general, it has the following requirements: (1) the fastest response time; (2) the smallest or no overshoot; (3) good rejection of disturbance. However, the system is very complex.

The PID controller and its improved form may satisfy (1) and (2) through an intelligent method, such as the fuzzy method, neural network method, and so on. However, in the mathematical form, it cannot eliminate the time-varying disturbance. The integral action can only handle the constant disturbance; the phenomenon can be proved mathematically. Meanwhile, in mathematical form, it is a linear combination based on error, and it is a known fact that the PID parameters, once adjusted, can be difficult to re-correct [5]. Therefore, we need a controller that is easy to implement and can satisfy the above three requirements. Han and Gao [13–17] constructed and developed the ADRC theory, which has been widely adopted in many industry areas. It mainly consists of a state feedback controller and an extended observer, which is the inheritance and development of the PID controller [21–24]. It retains the advantages of the PID controller but has a better robust performance. The stability theory has been proved in [25]. Based on the advantages of ADRC, we chose it as the controller of the spraying quantity.

**Remark 4.** *In the process of spraying, it requires the high performance of the adjustment time and steady-state error. At the same time, large overshoots will also cause pesticide waste in a short period time. Therefore, a cost function in the form of (10) is designed to solve the optimal parameters through PSO, corresponding to the $K_p$, $K_i$, $K_d$ in the PID controller and the $\omega_o$, $\omega_c$ in the LADRC controller.*

## 4. Simulation and Experimental Verification

### 4.1. Disturbance Measurement of Diaphragm Pump

In this study, the spray performance experimental platform (Figure 6) was used to measure the disturbance in the pipeline at different speeds. The pressure signal was selected as the measured value, and the speed of the diaphragm pump was set at 200, 240, and 320 rpm, respectively. Pressure data were collected through a data acquisition unit and median filtering was carried out [4]. The data waveform is shown in Figure 7. The mean values of pressure pulsation were approximately 0.24, 0.36, and 0.52 MPa, respectively, and the amplitude of pressure pulsation was approximately 0.05, 0.07, and 0.09 MPa, respectively. With the increase in the speed of the diaphragm pump, the pressure pulsation period became smaller.

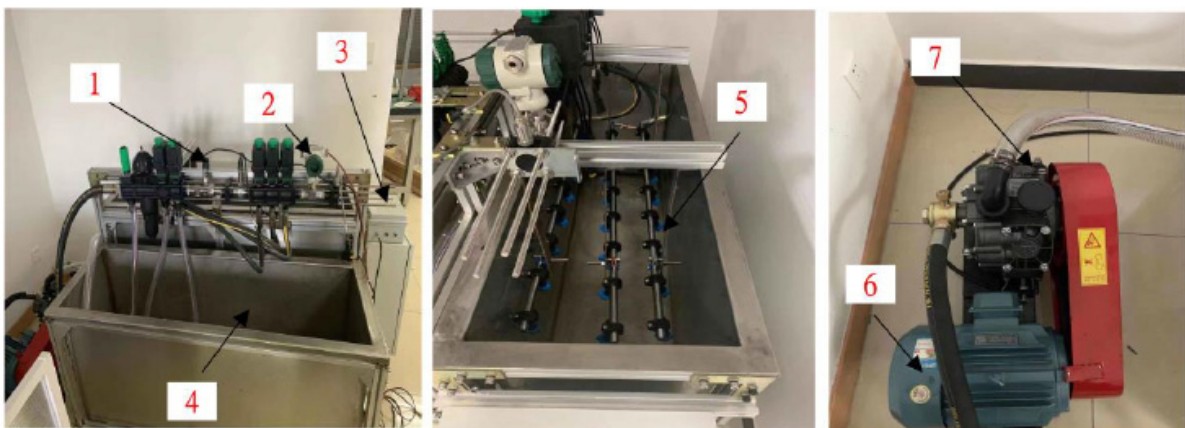

**Figure 6.** Spraying performance experimental platform: 1, pressure sensor; 2, flow sensor; 3, data acquisition unit; 4, tank; 5, nozzles; 6, motor; 7, diaphragm pump.

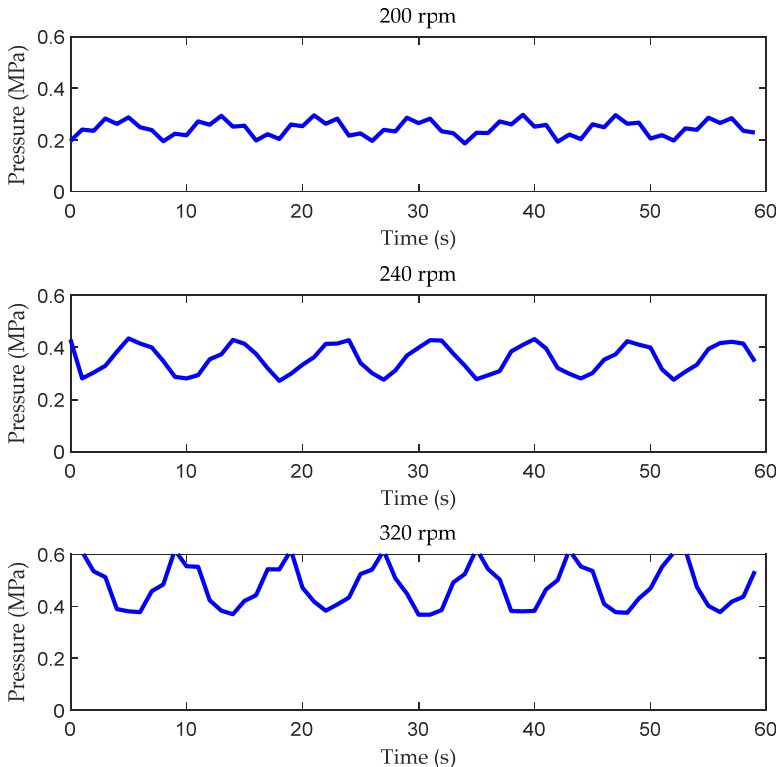

**Figure 7.** Pressure curves at different speeds.

**Remark 5.** *In the process of the application operation, the acceleration and deceleration of the vehicle will cause the change in the speed of the diaphragm pump, which will affect the internal flow of the pipeline. High precision and low-range flow sensors are expensive. In this study, the pressure pulsation in the pipeline was indirectly measured, and the flow disturbance caused by the change in speed of the diaphragm pump in the pipeline is explained by combining the pressure and flow approximate relationship.*

*4.2. Control Algorithm Simulation Analysis*

Based on the theoretical design of the PID and LADRC controllers in the second section, the simulation models of the three controllers were established by MATLAB simulation tools in this paper. The parameters of the PID controller, "PID + Filter" control, and the LADRC controller were tuned, respectively, by the PSO optimization algorithm according to the cost function index of Equation (13), and the optimal controller parameters under the condition of no disturbance being obtained. After tuning the parameters, we set $k_p = 0.28, k_i = 0.0602, k_d = 0.0427, b_0 = 22, \omega_o = 13, \omega_c = 42$, and the low pass filter time constant was set to 0.04.

The simulation results showed that the LADRC controller is significantly better than the traditional PID controller and the improved PID controller in terms of response time and overshoot under the condition of no disturbance, where the overshoot could be ignored and the steady-state error was less than 0.01%. The overshoot of the traditional PID controller and the improved PID controller is more than 0.2% and the steady-state error was less than 1%. The simulation result is shown in Figure 8.

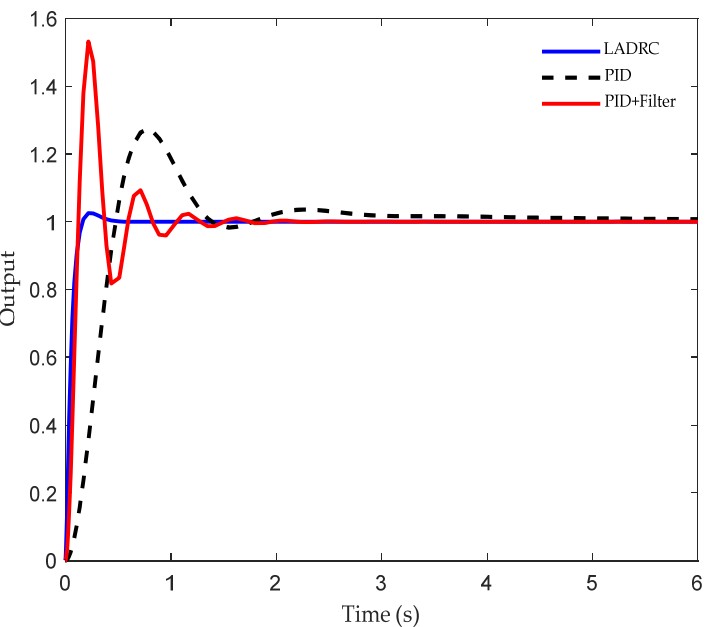

**Figure 8.** Step response simulation results without disturbance.

In order to compare the disturbance suppression ability of the three controllers, we carried out a simulation comparison under a constant disturbance. The simulation result is shown in Figure 9. Compared with the traditional PID controller and the improved controller, the LADRC can significantly reduce a constant disturbance. The LADRC controller can compensate the disturbance totally and quickly through the LESO, hence the disturbance immunity of the LADRC is better than an improved PID controller.

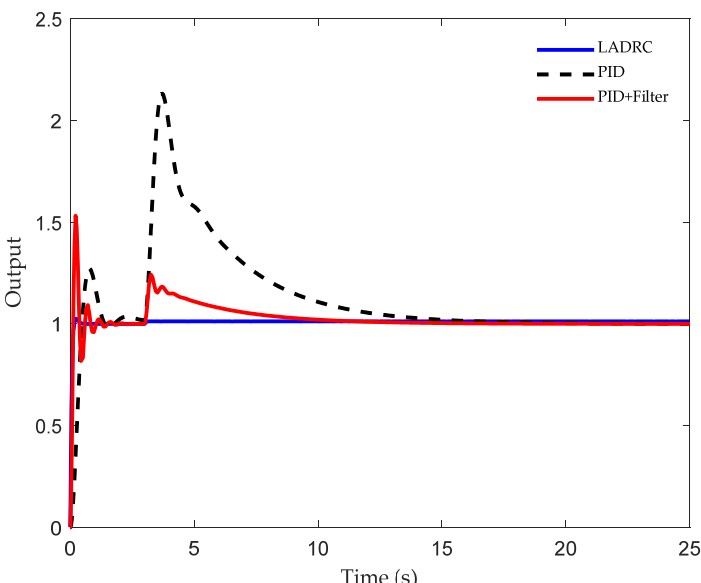

**Figure 9.** Step response simulation results with constant disturbance (under three controllers).

This study assumed that the system is subject to slope disturbance in the acceleration or deceleration stage and sinusoidal disturbance in the uniform speed stage, according to the measurement results of pressure values in the pipeline at different diaphragm pump speeds in Section 3.1. In this case, a severe disturbance was used to compare the robustness

of the three controllers, $d = \begin{cases} 0.3t & t < 5 \\ 1.5 & 5 \leq t < 10 \\ -0.2t + 3.5 & 10 \leq t \leq 15 \\ 0.5\sin(t) & t > 15 \end{cases}$. In the case of disturbance,

in the LADRC controller, the LESO is used for the real-time disturbance observation, and the controller compensates for the disturbance through the disturbance observer. The simulation results are shown in Figures 10 and 11. The results show that the linear extended state observer can effectively observe the total disturbance after 0.77 s.

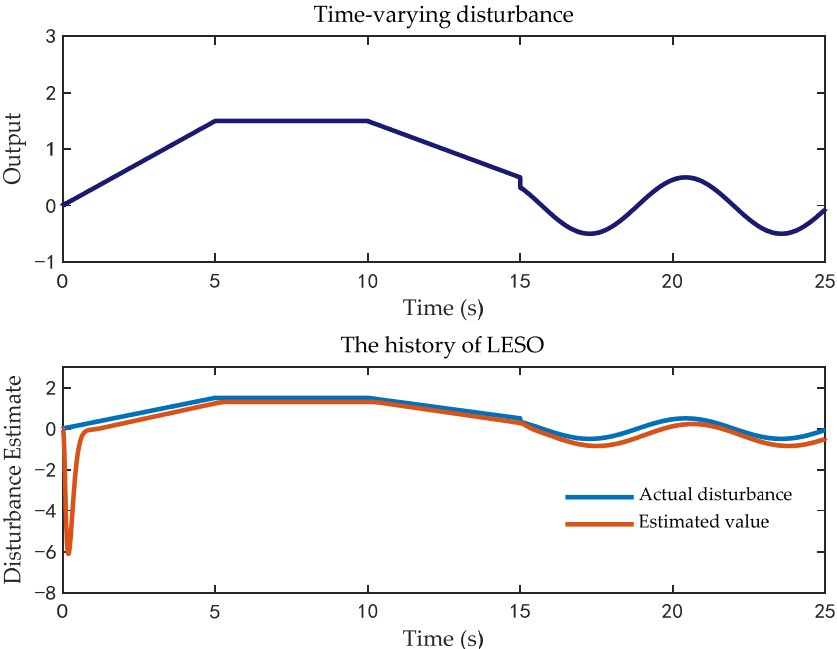

**Figure 10.** Simulation results of disturbance observer.

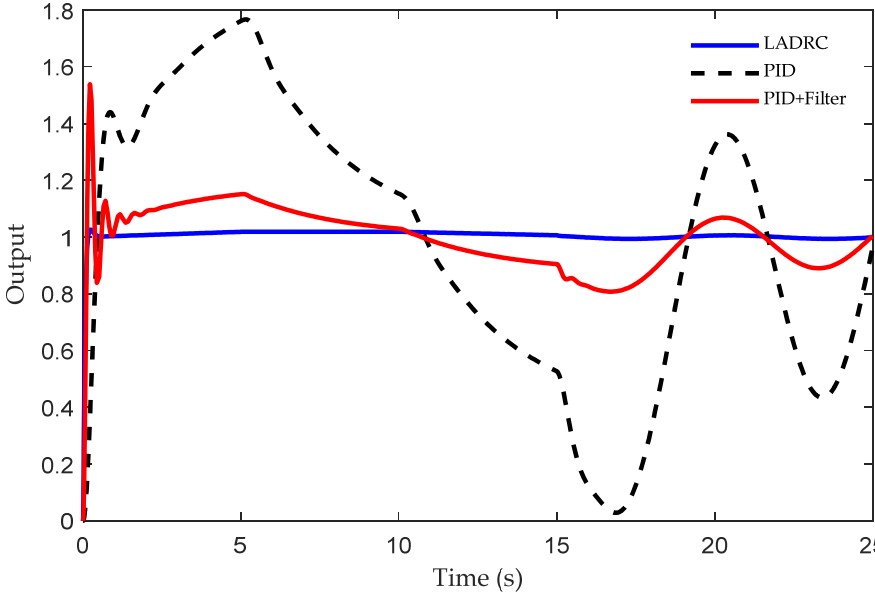

**Figure 11.** Step response simulation results with disturbance.

The simulation results show that under the condition of disturbance, the traditional PID controller and the improved PID controller have a large overshoot and steady-state error, as shown in Figure 11. The LADRC controller used in this study can compensate for

the time-varying disturbance in real-time and shows good robustness. Due to the integral action, the traditional PID controller and the improved PID controller cannot handle a time-varying disturbance. Due to the low pass filter in series, the improved PID is better than a traditional PID.

In Section 2.1, the controlled plant of the transfer function model was set up by the response curve method in order to obtain the nominal system model, but the phenomenon that the controlled object parameter is uncertain exists. This study in step simulation without disturbance in the simulation of 10~13.5 s time will add 20% to the nominal system model parameter perturbation. The simulation results are shown in Figure 12.

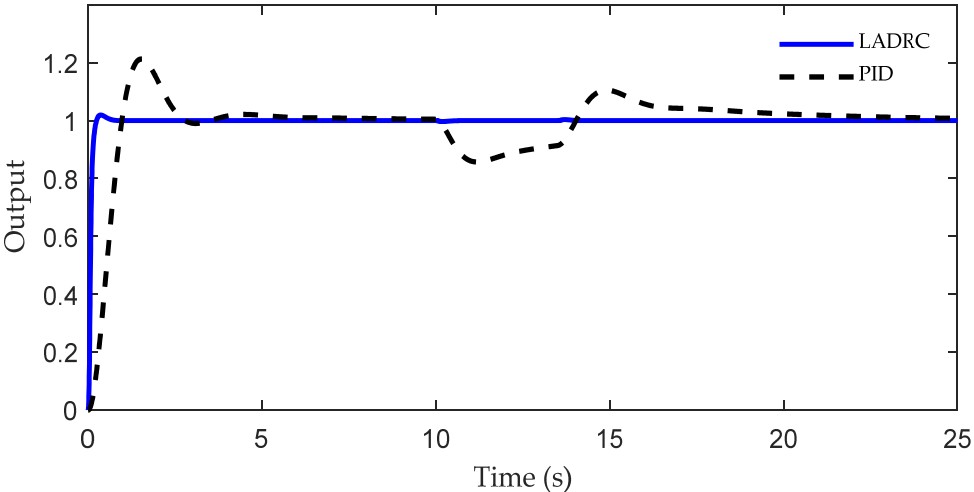

**Figure 12.** Step response simulation results with parameter perturbation.

As shown in Figure 12, in the case of model parameter perturbation, it takes approximately 12 s for the traditional PID controller to reach a steady-state, while the LADRC controller returns to the steady-state in less than 1 s. Therefore, both the external disturbance and internal disturbance suppression of the LADRC controller are significantly better than a traditional PID controller.

### 4.3. The Field Test

The experiment was carried out in the experimental site of Zhenjiang Runguo Agricultural Development Co., Ltd. (Zhenjiang, China). First, the vehicle was calibrated at 6, 8, and 10 km/h field speeds. The field spraying experiment was carried out according to the calibrated position, and the target spraying flow rate was set as 100~150 L $hm^{-2}$. Before the operation, relevant operation information was input on the spraying control interface. The operation parameter monitoring system collected the flow data in the pipeline once a second and converted it into the actual spraying quantity. The results are shown in Figure 13 and Table 1.

To better fit the actual situation, the experiment carried out the stability control test of 6~8 km/h and the 8~10 km/h accelerated motion test. Similarly, the operation parameter monitoring system collected the flow data in the pipeline once per second and converted it into the actual spraying quantity. The data collected under the two conditions are depicted in Figure 14.

The experimental results show that the response speed and steady-state error of the LADRC controller are significantly better than the traditional PID controller at three operating speeds under the condition of constant speed. Compared with PID, the LADRC response speed is reduced by approximately 3~5 s, the steady-state error accuracy is increased by 2~9%, and the difficulty of the tuning control parameters is greatly reduced. In the case of the accelerated motion test, the system has a large steady-state error under the action of the PID controller, but the LADRC controller can achieve the preset value

within a lesser time. The experimental results show that the LADRC controller has the advantages of strong robustness, easy parameter tuning, and weak model dependence, and it also indicates that the "observer + controller" control framework is suitable for spraying quantity stability control.

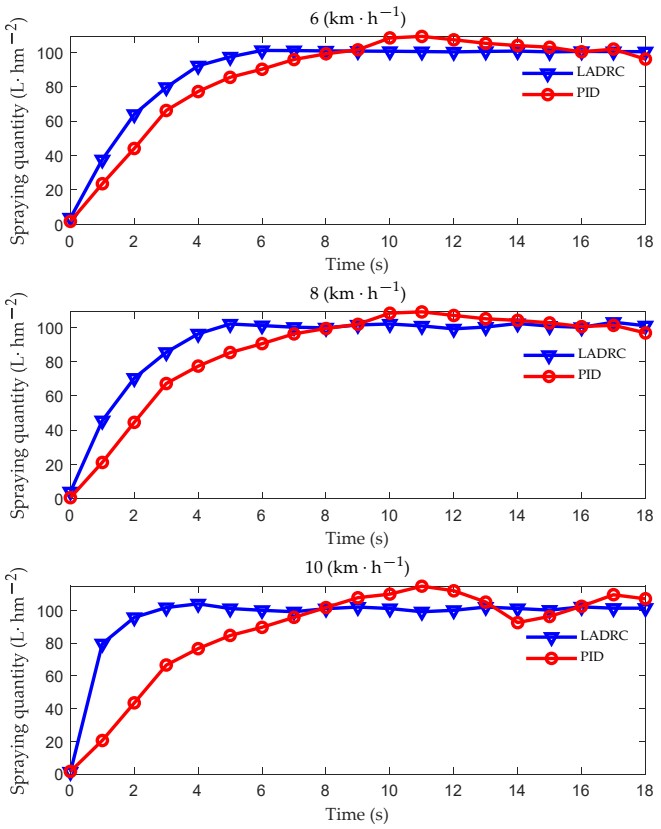

**Figure 13.** Field spray quantity data curve (uniform motion).

**Table 1.** Spray test results.

| Test Group | The Set Rate/ (L hm$^{-2}$) | Speed (km h$^{-1}$) | The Actual Rate (L hm$^{-2}$) | Error % |
|---|---|---|---|---|
| LADRC | 100 | 6 | 100.2 | 0.2 |
|  | 100 | 8 | 101.8 | 1.8 |
|  | 100 | 10 | 103.7 | 3.7 |
|  | 125 | 6 | 124.1 | −0.9 |
|  | 125 | 8 | 127.3 | 2.3 |
|  | 125 | 10 | 128.4 | 3.4 |
|  | 150 | 6 | 147.4 | −2.6 |
|  | 150 | 8 | 153.5 | 3.5 |
|  | 150 | 10 | 153.2 | 3.2 |
| PID | 100 | 6 | 103.2 | 3.2 |
|  | 100 | 8 | 104.5 | 4.5 |
|  | 100 | 10 | 109.2 | 9.2 |
|  | 125 | 6 | 121.8 | −3.2 |
|  | 125 | 8 | 119.4 | 5.6 |
|  | 125 | 10 | 121.4 | −3.6 |
|  | 150 | 6 | 154.5 | 4.5 |
|  | 150 | 8 | 157.4 | 7.4 |
|  | 150 | 10 | 142.2 | −7.8 |

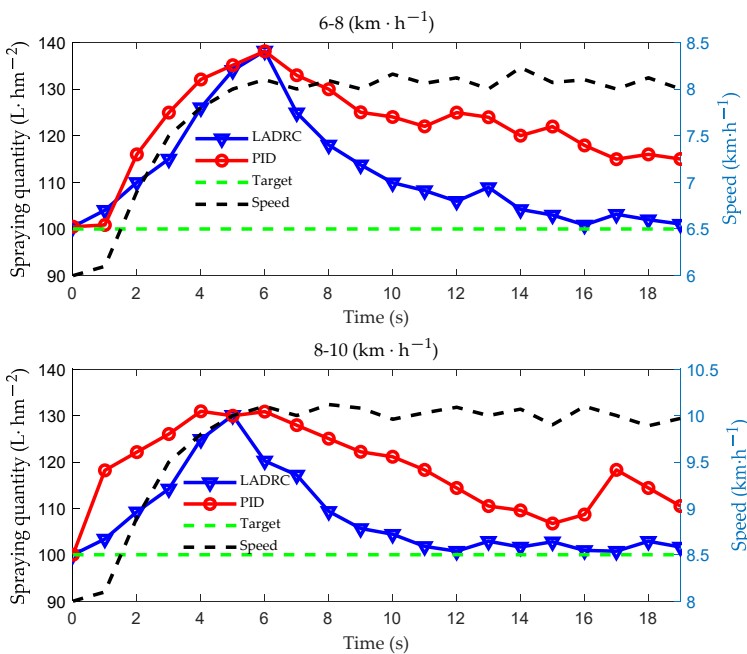

**Figure 14.** Field spray quantity data curve (accelerated motion).

The same results of the PID controller in spraying quantity can be obtained in [4–8]. In the process of acceleration, the speed of the diaphragm pump changes. It will cause the increase in flow quantity in the pipe and, meanwhile, will produce the time-varying disturbance, which has often been ignored in [4–8]. On the other hand, the process of acceleration can be regarded as a constant disturbance. As a result of the constant and time-varying disturbance, it is difficult for the traditional PID controller to achieve good results. In the past, the literature [4–8] mainly focuses on the tuning of PID control parameters, which relies on the action of the integral to suppress the disturbance. Formally, the PID controller is a linear combination of error and cannot eliminate disturbances actively [12–16]. LADRC can estimate the lump disturbance quickly and accurately. It can eliminate the disturbance through feedforward control [12–16]. Hence, the results of the LADRC controller are better than the traditional PID controller.

## 5. Conclusions

In this paper, a LADRC controller was adopted in order to adjust the spraying quantity under both time-varying disturbance and uncertainty. LADRC can improve the dynamic performance and the control accuracy. Meanwhile, a PSO algorithm was used to find the best optimal controller parameters.

We conducted the spraying quantity control simulation under three controllers. The simulation results of this study show that the LADRC controller shows better disturbance immunity and robustness than the traditional PID controller and the improved PID controller. To verify the actual performance, we built a prototype and tested it in the field. The experimental results show that the response speed and steady-state error of the LADRC controller are significantly better than the traditional PID controller.

Compared with PID and the improved PID controller, the LADRC response speed is reduced by approximately 3~5 s, the steady-state error is reduced by 2~9%, and the difficulty of setting the control parameters is greatly reduced. We conclude that the LADRC controller is suitable for the spraying quantity control and can also improve the performance. More studies will be carried out in the future.

**Author Contributions:** Conceptualization, X.J. and X.W.; methodology, X.J.; software, X.J.; validation, X.W.; formal analysis, X.J.; investigation, X.J.; resources, X.W.; data curation, X.W.; writing—original draft preparation, X.J.; writing—review and editing, A.W.; visualization, A.W.; supervision, X.W.;

project administration, X.W.; All authors have read and agreed to the published version of the manuscript.

**Funding:** This research was funded by the "National Key Research and Development Program (2016YFD0700104)", the "Key Research and Development Program of Jiangsu Province (BE2018331)", and the "Key Research and Development Program of Jiangsu Province (BE2019318)".

**Conflicts of Interest:** The authors declare no conflict of interest.

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
