# Peer review of "Precision Control of Spraying Quantity Based on Linear Active Disturbance Rejection Control Method"

_agriculture, doi:10.3390/agriculture11080761_

Round 1
Reviewer 1 Report
The authors addressed most of the comments but some are not. and the audthors didn't reply to the comments I made earlier to justify their reply.
I will add the comments again:
- Modeling and problem description
Figure 1. Shouldn’t part no. 2 be a shaft? It doesn’t have any thread to be considered a screw.
Why was the controlled plant gain chosen as 71.4, The first-order
inertia time constant of the controlled object was 4.6 s, and the delay time constant was 0.8 s?
Figure 3. Add more details to sub-figure a, No need for sub-figures d, subfigures e and f are not in English, so you either have the GUI in English or delete this sub-figure. Names of parts in teh figure are not clear. Change the font color.
Figure 12. What is the purpose of this figure? It doesn’t add any details. You can add a video as a supplementary material.
Figure 12. Where is caption? Improve the resolution of axis.
Figure 13. Improve the resolution of axis.
Figure 14. Improve the resolution of axis.
Reviewer 2 Report
The paper is significantly improved. Authors have improved their paper taken into consideration the remarks from the reviewers. You can see my extra comments in the following.
1. I believe that the Introduction is a bit short. I would appreciate if you could elaborate on the existing spraying quantity control and their quality of results (measured error from the control model vs real figures using references). In addition, in the same context you could provide a better outlook of the intelligence method (fuzzy logic or neural networks) applications on the subjects of spraying quantity control. Please extend the review to intelligence systems applied to spraying quantity control applications. After giving an overview of different cases spraying, you might also point out the papers produced in this subjects for certain plant or others. This way, you will provide evidence to the reader that your work is unique and original (which I totally believe so).
2. Line 336, 349. Please do not start paragraphs with a sentence that just says that a Fig. shows specific information - you might use the main text to state what you have found in your study, and then cite a Fig that provides illustrative detail.
3. Fig. 6. - Generally it prefer Figs to define terms, symbols or lines etc. in the caption, not the Fig itself.
4. The format of Figures and tables should be consistent, including type of font, axis unit, etc.
5. Section 4.3. - If you do not discuss the results in the same section, you will need to separate them! whatever you choose to do, you will need to use references that explain your work results!
6. Please add a discussion section and integrate it in the section 2 and 3 in the appropriate parts for each of the remarks given. It should be noted that you need to use references to explain your results and discuss them when you are trying to explain something found and not presented in previous work.
7. References needs improvement based on conference format. Names of authors should be abbreviated, etc. The format of literature review [12, 22] should be consistent, such as the lower-case and capital letter of paper title.
Author Response
Please see the attachment

This manuscript is a resubmission of an earlier submission. The following is a list of the peer review reports and author responses from that submission.
Round 1
Reviewer 1 Report
agriculture-1273491-peer-review-v1
The study looks promising in presenting and verifying a new technique to control the spraying operation. The author provided enough explanation of the materials and methods as well as the results. With some modifications in the introduction, conclusions, and also figures quality.
Abstract
Line 11: Delete and so on. This is not a scientific word.
Line 35: “Guzman et al.”- Add the date of publishing or the format the citation to fit the journal standard. Same issue in Line 45. Review and fix similar issues.
- Introduction
The authors need to add more details in terms of the available commercial controllers. It will be good to add a figure that explains the PID operation idea in general.
- Modeling and problem description
Figure 1. Shouldn’t part no. 2 be a shaft? It doesn’t have any thread to be considered a screw.
Why was the controlled plant gain chosen as 71.4?
Line 145: Change “vehicles” to vehicle.
Figure 3. Add more details to sub-figure a, No need for sub-figures d, subfigures e and f are not in English, so you either have the GUI in English or delete this sub-figure.
Figure 5. Improve the resolution.
Figure 6. The names of different parts are not clear.
Figure 7. Improve the resolution.
Figure 9. Add a sub-title for each subfigure. The legend of the bottom sub-figure is not clear.
Figure 12. What is the purpose of this figure? It doesn’t add any details. You can add a video as a supplementary material.
Figure 13. Improve the resolution.
Conclusions
The authors need to state the conclusions in a better way. You need to add brief details such that if the conclusions section is the only part read ion the paper, it will be enough to understand what has been done and what is the output. Additionally, the future work should be listed in a more convincing way.
Reviewer 2 Report
The target audience for this paper is clearly the agricultural community. As optimal control is not common in this field a good clear overview of what feedback control with PSO actually is vital in a paper like this. There are several areas where the writing needs a bit of work or is unclear. The point I perceive as most important from this paper is to reduce the distribution for spraying application and its flow control value required for regulating the flow of liquid in the main ripe. This is an important topic, which should be further addressed in research. Nevertheless, this study is still at its initial stage which is more like a conference paper. There seems to be much work to be done.
Major Concern
- Do you consider the effect of external disturbance in the system? There would need to be some clear demonstration that your implementation of what may be a similar system model for precision control of spraying quantity does do more than other transfer function model.
- The next major problem is notation. This is extremely confusing and appears to change from section to section. For example : the dimension of the x, y, and w.
- I am also concerned about your definition of disturbance. Disturbance is sometimes difficult to pin down in a testable way, and your definition is not unreasonable, but I cannot see how it relates to any specific information of " field speeds " or “flow data” (line 301).
Methodology -
- The author does not depict the detailed mathematical inference process, how to avoid disadvantages of the SPO algorithms, and how to divide or switch the distribution during spraring? The authors do not explain the proposed control method how it can be used although the flow chart of PSO algorithm is shown in Fig. 5. The deduction process of proposed method is too rash (section 3).
- Line 205-206: you say “To solve the problem that the control parameters of the traditional controller are difficult to tune” What is traditional controller? This sentence needs to be clarified. The illustration of Figure 5 need to be enhanced.
- On lines 314-316 you say " the LADRC controller has the advantages of strong robustness, easy parameter tuning, and weak model dependence " and this does not allow the reader to investigate the basis. Line 112: What is difference between disturbance and uncertainty in your system? Content of Section 2 is unclear. Please clarify the definition of f. How is it related to the d (line 266)?
- How to use PSO method to calibrate the most sensitive parameter values to improve the robustness of the system model (LESO)?
Simulations/ Experiments -
- One thing that appears to me to be lacking is the comparison with the state of the art. While the authors do mention PID approaches briefly, as I see it the most effective way of dealing with disturbance in spraying control is to implement a simple adaptive PID filter. This is very computationally efficient, and very effective in eliminate disturbance. How would the proposed method compare to such a simple solution?
- The experiment process of field test in not clear. The process of sprayer setup and flow rate measurement do not illustrated in this paper. Experiments and measurement results is roughly. Several of the figures (Figs. 3d, 3e, and 12) accomplish nothing and should be presented differently. How to use least squares identification method to calibrate the most sensitive parameter values to improve the robustness of the model?
- The depiction of spec. (or brand) of all sensors and controller is required.
Conclusions -
In my opinion, more scenario simulations should be taken to testify the conclusion. I would like to see a more in depth description of the novelty of this work, and how it relates to previous work in the field of precision control of spraying quantity.